# Neurodegenerative Diseases: Implications of Environmental and Climatic Influences on Neurotransmitters and Neuronal Hormones Activities

**DOI:** 10.3390/ijerph191912495

**Published:** 2022-09-30

**Authors:** Emmanuel A. Ayeni, Ahmad M. Aldossary, Daniel A. Ayejoto, Lanre A. Gbadegesin, Abdullah A. Alshehri, Haya A. Alfassam, Henok K. Afewerky, Fahad A. Almughem, Saidu M. Bello, Essam A. Tawfik

**Affiliations:** 1Chengdu Institute of Biology, Chinese Academy of Sciences, Chengdu 610041, China; 2University of Chinese Academy of Sciences, Beijing 100049, China; 3National Center of Biotechnology, Life Science and Environment Research Institute, King Abdulaziz City for Science and Technology (KACST), Riyadh 12354, Saudi Arabia; 4Department of Industrial Chemistry, University of Ilorin, Ilorin 240003, Nigeria; 5Institute of Mountain Hazards and Environment, Chinese Academy of Sciences, Chengdu 610041, China; 6KACST-BWH Center of Excellence for Biomedicine, Joint Centers of Excellence Program, King Abdulaziz City for Science and Technology (KACST), Riyadh 12354, Saudi Arabia; 7Department of Neurobiology, School of Basic Medicine, Tongji Medical College, Huazhong University of Science and Technology, Wuhan 430074, China; 8School of Allied Health Professions, Asmara College of Health Sciences, Asmara P.O. Box 1220, Eritrea; 9Institute of Pharmacognosy, University of Szeged, 6720 Szeged, Hungary

**Keywords:** neurodegenerative diseases, hormones, neurotransmitters, environment, climate, public health

## Abstract

Neurodegenerative and neuronal-related diseases are major public health concerns. Human vulnerability to neurodegenerative diseases (NDDs) increases with age. Neuronal hormones and neurotransmitters are major determinant factors regulating brain structure and functions. The implications of environmental and climatic changes emerged recently as influence factors on numerous diseases. However, the complex interaction of neurotransmitters and neuronal hormones and their depletion under environmental and climatic influences on NDDs are not well established in the literature. In this review, we aim to explore the connection between the environmental and climatic factors to NDDs and to highlight the available and potential therapeutic interventions that could use to improve the quality of life and reduce susceptibility to NDDs.

## 1. Introduction

Neurodegenerative diseases (NDDs) are neuronal diseases that affect the central and peripheral nervous systems (CNS and PNS) of the brain’s composition and function. NDDs can lead to progressive loss of nerve structure needed for brain functions [1]. Neurodegenerative disease is one of the significant causes of mortality and disability globally [2,3]. Prominent NDDs include Alzheimer’s disease (AD), Parkinson’s disease (PD), Huntington’s disease (HD), amyotrophic lateral sclerosis (ALS), multiple sclerosis (MS), psychological disorders, motor neuron disorders, dementia with Lewy bodies (DLB), vascular dementia (VaD), epilepsy, cerebral ischemia, behavioral disorders and mental illness [4]. NDDs are associated with genetics, cell signaling impairments, neuronal apoptosis, inflammatory response, deposition of aggregated proteins, mitochondrial dysfunctions, oxidative stress, aging, gender, mutations, ethnicity, environmental exposures, and climate [1,5,6,7,8,9,10,11].

AD is a result of low acetylcholine in the brain. The pathological features of this condition involve the deposition of β-amyloid in the extracellular surface of neurons and the neurofibrillary tangles arising from the intracellular accumulation of hyperphosphorylated Tau protein. Some proteins are implicated in the progression of AD, such as the amyloid precursor protein (APP) and presenilin (PSEN 1 and 2), which lead to the early onset of AD [12].

PD affects various motor and non-motor neurons. It is characterized as a gradual reduction of dopamine (DA) level in the brain’s nigral striatum, leading to postural imbalance, uncontrolled body gaits, bradykinesia, muscle stiffness, and resting tremors. PD prevalence increases with age [13]. Pathophysiological features of PD include the accumulation of aggregated alpha-synuclein (SNCA) in the intraneuronal cytoplasm known as Lewy bodies (LB) and the progressive damage of DA neurons in the substantia nigra pars compacta. Mutations in the Leucine-Rich Repeat Kinase 2 (LRRK 2), Parkin RBR E3 ubiquitin-protein ligase (PARK 2 and 7), PTEN-induced putative kinase (PINK1), and SNCA genes are known to cause familial PD, and some complex interaction in the gene variants could also lead to PD sporadically [14].

HD is an inherited neurodegenerative disorder, with symptoms including psychological changes, cognitive deterioration leading to dementia, and symptomatic motor changes, such as chorea and dystonia [15]. These symptoms are correlated with selective degeneration of the striatal and cortical neurons [16]. HD is caused by abnormal extension of cytosine, adenine, and guanine (CAG) repeat code for a polyglutamine (poly Q) tail in the huntingtin gene [17]. There are currently no therapies to prevent the onset or slow down HD progression, and the mechanism involved is unknown.

Psychological disorders are related to mental retardation; this neurological impairment affects cognitive behavior, mood, and schizophrenic diseases [18]. Mental retardation is linked to GABA, low DA, and serotonin [19].

Many people globally suffer from different NDDs [20,21,22,23,24]. For instance, AD and PD patients have increased significantly more than two-fold since 1990 [5]. To date, NDDs has no permanent cure, affecting the quality of livelihood, and healthy living is lost to years of illness and even death. At the same time, NDDs pose a significant burden on the socioeconomic and health of caregivers, including emotional instability, depression, and quality of well-being [25,26]. Global burden diseases (GBD) are associated with environmental factors, such as pollution, heavy metals, pesticide use and ingestion, detergents, chemicals and solvents, and other industrial bio-products in several NDDs [27,28,29,30]. These environmental factors could modify and trigger psychological and mental disorder conditions in the brain by crossing the blood-brain barrier (BBB) and disrupting the functions of the CNS and PNS [31].

On the other hand, climatic conditions resulting from global warming, destroying ecosystems and natural habitats, have been directly and indirectly linked to NDDs. Cheshire stated that climate change which masks the body’s ability to regulate temperature, could cause mental disorders [7]. A climate change report projected additional 250,000 deaths per year between 2030 and 2050 and a potential 38,000 deaths from heat exposure and associated neurological conditions among elderlies [32,33,34].

Neuronal hormones such as acetylcholine (ACh), DA, and serotonin (5-HT) control the physical and psychological functions of the brain, and their depletion could lead to various forms of NDDs symptoms such as locomotion disorder, tremor, body rigidity, bradykinesia, memory dysfunction, and cognitive defects. Neurotransmitters are chemical messengers that transmit signaling molecules secreted by neurons across synapses to different target cells [35]. They share information within the brain and communicate with other body parts. Neurotransmitters interact through cell surface receptors and regulate several ion channels, thereby modulating cell signals across synaptic cells between neurons, muscles, or gland cells [36]. Neurotransmitter activity affects the excitatory and inhibitory processes and may lead to NDDs.

Neuronal drugs and interventions for managing NDDs focus on symptoms and short-term relief episodes. These drugs are accompanied by several side effects such as toxicity, low bioavailability, addiction, co-administration, neuro-inflammation, and higher doses regimen that could worsen diagnosis and deteriorate brain capability [37]. The non-pharmacological approaches, such as body exercises, running, physical activity, and emotional stability, are beneficial to enhancing the quality of life in patients with several types of neurological diseases [38,39], especially at the onset of the neurological diagnosis. Still, a lasting treatment would be desired to deflate the burden and brings hope to the management of NDDs. Therefore, this review discusses the roles of major neuronal hormones and neurotransmitters in the pathogenesis of NDDs and highlights the influence of environmental and climatic factors on their activities. Another potential aim of this review is to highlight the pharmacological and non-pharmacological strategies that could be implemented to control the NDDs progression triggered mainly by environmental and climatic factors. This can be achieved via defining the pathophysiological mechanisms of the NDDs and their relation to environmental and climatic changes. Hence, the pharmacological and non-pharmacological strategies for NDDs management will be explained accordingly to link each strategy with the appropriate environmental or climatic factors. Since controlling environmental and climatic factors that affect the neurotransmitters’ stability or hormones activity and promote the progression of NDDs is not always feasible; thus, the pharmacological and non-pharmacological approaches will be discussed as crucial NDDs management strategies.

## 2. Pathogenesis of Neurodegenerative Diseases

The pathogenesis of NDDs is inadequate and unclear. However, several studies attributed NDDs to brain injuries, genetics, stress, mutations, environment, lifestyle, and diet [40]. The world health organization (WHO) stated that over 50 million living with NDDs worldwide [41]. NDDs increase with age, and dementia is notable among older people [42,43]. Pathophysiological features of NDDs could be attributed to the misfolding and depositions of some aggregated proteins; for instance, PD is linked to the misfolded α-synuclein. In distinction, amyloid-β (Aβ) aggregation and neurofibrillary tangles are linked to hyperphosphorylated *tau* in AD, mutated huntingtin (HTT) in HD, and TAR DNA-binding protein 43 (TDP-43) in ALS [44,45,46,47]. These factors could also be linked to early life exposure to some environmental conditions, cardiovascular diseases, diabetes, cancers, and other immunopathological disorders [40,48].

AD and PD are common forms of NDDs, and both share interrelated symptoms and clinical signs such as body rigidity, bradykinesia, memory loss, depression, tremor, and related psychological disorders. AD is associated with neuronal cell death between the hippocampus and cortex regions of the brain. The symptoms are related to forgetfulness, memory impairment, and cognitive dysfunction leading to death [49,50,51]. The pathophysiological activities of AD are not yet fully understood, but several studies suggested that the accumulation of neurotoxicity induced by misfolded Aβ, neurofibrillary tangle accumulation, and phosphorylated tau proteins, alongside other genetic and clinical evidence described AD [52,53,54,55]. PD is a progressive neuronal disorder resulting from the loss of DA in the substantia nigra pars compacta and the degeneration of projecting nerve fibers in the striatum [56]. Many in vivo and in vitro studies associate PD with stress and mitochondrial dysfunction. The increase in reactive oxygen species (ROS), nitric oxide (NO), and superoxide could lead to neuroinflammatory responses triggered by several events that can predispose NDDs and even death [57,58]. Furthermore, the activated microglia produced by these responses could affect DA neurons such as α-synuclein, neuromelanin, matrix metalloproteinase-3, and other proteins linked to familial forms of PD [59,60,61].

It is well known that CNS inflammation and immune activation play a significant role in the pathophysiology of the NDDs [62]. Hrelia et al. stated that redox signaling dysregulation contributes to several age-related diseases and is responsible for endothelial dysfunction in most pathophysiological neurodegenerative conditions [55]. Other studies linked NDDs to neuroinflammation, immune response, immunological cell types, and cytokine [63,64,65]. Tuppo and Arias reported that AD is associated with neuroinflammation and increased activation of the immune system and that accumulation of amyloid proteins increases toxicity in the brain, which can trigger neuronal diseases [66]. Furthermore, the BBB could protect the CNS from immune activation. Still, Fakhoury noted that BBB could be more permeable during inflammation, rendering the brain vulnerable to bacterial and virus infections [67].

### 2.1. Role of Neurotransmitters and Neuronal Hormones in the Pathogenesis of Neurodegenerative Diseases

Neurotransmitters are endogenous chemicals that allow communication between neurons and contribute to daily activities and functions of the body [35]. They are diverse and classified based on chemical composition and processes in the CNS and PNS [68,69,70]. Deterioration occurs due to depletion in the neuronal hormones, leading to cognitive dysfunction [71]. Neurotransmitters are also involved in metabolic disorders, such as diabetes, associated with insulin deficiency which affects the carbohydrate metabolism linked with various CNS and PNS [72]. Neuronal hormones such as ACh, DA, 5-HT, norepinephrine (NE), melatonin, and epinephrine (EPI) have been implicated in NDDs. Neurotransmitters such as glutamate, gamma-aminobutyric acid (GABA), histamine, thyroid hormone, tyrosine, glycine, NO, and hydrogen sulfide (H_2_S), among others, contribute to the operational metabolism of neuronal cells. They are critical factors in neurological and NDDs of the CNS [73]. Table 1 summarizes the most common neuronal hormones and neurotransmitters associated with the pathogenesis of NDDs.

### 2.2. Implications of Environment and Climate Influences on Neurotransmitters and Neuronal Hormones Activities

Global environmental degradation and climate change undeniably threaten human health, water bodies, and animals. Studies suggest overwhelming evidence of their impact as a public health concern and global burden of diseases. Environmental exposures and unfavorable climatic could initiate inflammation, temperature variability, and heat stress in NDDs. During these processes, several neurotransmitters and neuronal hormones are lost and depleted. For example, oxidative stress, heavy metals, and environmental pollution have been reported to have significantly reduced ACh and DA in AD, PD, and VaD. Other neurotransmitters and hormones such as 5-HT, tryptophan, glutamate, GABA, NE, and corticotrophin-releasing factor also significantly decreased [100,101]. However, the amount of damage contributed by the environment and climate to neurotransmitter degradation or hormonal depletion is unknown. Generally, oxidative stress in human activities contributes to behavioral and metabolic fluctuations, such as mental stress from daily activities contributing to the loss of these neurotransmitters. Mental stress and low 5-HT levels are among the most identified mood disorders, such as major depression disorder (MDD) and anxiety disorders [102,103]. Depletion levels of 5-HT or activity increase in response to stress. Ciobica et al. used experimental rats to understand the effect of 5-HT depletion using 5, 7-DHT lesions of the hypothalamic paraventricular nucleus and how it affects behavioral performance via interactions. The results showed no significant change using 5,7-DHT lesions but increased neuronal oxidative stress [104].

Further, tryptophan depletion in the brain could lead to a severe acute depressive mood or anxiety disorder [105]. Zhang et al. explored common changes induced by acute mental stress (AMS) and acute tryptophan depletion (ATD). They confirmed that AMS and ATD cause changes in common neural pathways, which might mark susceptibility to depression and anxiety as an early indicator of the NDDs [106]. In the study by Aguado-Llera *et al.*, tripartite of glycine-proline and glutamate (neurotransmitters) exerted a protective effect against Aβ on the somatostatinergic system by reducing the inflammatory environment that may activate different pro-survival pathways in the brain neurons. These neurotransmitters showed effects on learning and memory processes and might be potential therapeutics in developing anti-AD medications [107].

The role of the environment on several neurotransmitters can further be examined in the context of specific processes influencing NDDs, such as inflammation and aging. Bektas et al. linked aging with inflammation and the roles played by the environment, especially among the elderlies, due to weakened neurons [108]. Contributing factors such as environmental pollutants could also affect neuronal hormones. Several studies have implicated mercury, arsenic, manganese, lead, pesticides, biotoxins, persistent organic pollutants, and disruptive endocrine chemicals as possible agents of neuronal hormone depletion [109,110,111,112,113]. Thambirajah et al. reviewed the role of contaminants and environmental influences on thyroid hormone and how the ecological threats posed by these contaminants are further exacerbated by changing environmental conditions such as temperature, photoperiod, and ultraviolet radiation [114]. The environment positively or negatively impacts human mental health, which can shape the development and maturity of the brain and its functions [115]. There has been a significant increase in evidence of social influences and their long-lasting impacts on the brain circuits and human behavior, much like the conventional environmental exposures as toxicological or nutritional factors [116,117].

According to Aumann, environment and activity-dependent DA neurotransmitter plasticity in the adult substantia nigra showed imbalances in midbrain DA. It caused symptoms associated with several major behavioral disorders such as schizophrenia, addiction, obsessive-compulsive disorder, depression, PD, and attention-deficit and hyperactivity disorder [118]. Midbrain DA neurotransmitter plasticity may play an essential role in the etiology of these symptoms and offer new treatment options. Aumann et al. showed the first evidence of an association between environmental stimuli (photoperiod) and the number of midbrain DA neurons in humans. They suggested lower DA and DA signaling in winter compared with summer. These investigations also implicated that climate change contributes to neuronal hormone depletion through seasonal variations and could offer either pro-neurotransmitters benefits or anti-neuronal hormone deficit [119]. Dunlop and Nemeroff stated a strong association between low DA and depression symptoms in winter due to decreased DA signaling and a smaller number of DA neurons in winter [120]. We opined that precise temperature conditions, contributing ion channels, active conductance, and biochemical function parameters should be investigated on essential neuronal hormones and transmitters to ascertain the effects of environment and climatic factors and the best suitability approaches.

At the same time, neurotransmitter concentrations in affected individuals’ brains are believed to be influenced by natural elements such as cadmium [121]. Zatta and Frank identified the role of copper in brain development since they have the potential to cross the BBB, and the transportation and distribution are controlled by ATP7A in glia and neurons and thus play a significant role in the biosynthesis of neurotransmitters, such as DA and NE. Therefore, introducing valuable metals such as copper and zinc could reveal intrinsic findings in many NDDs [122]. A complex mixture of air pollutants, including lead, ozone, polycyclic aromatic hydrocarbons (PAHs), and delicate particulate matter in urban environments, particularly in many metropolitan cities, could negatively impact neuronal hormones and neurons. Short and long-term exposure to air pollution is associated with an increased risk of stroke, hidden infarctions, and brain shrinkage [123]. The degree of prenatal exposure to PAHs is also correlated with decreased brain white matter volume, cognitive decline, and a rise in indicators of attention-deficit/hyperactivity disorder [124]. Animal studies on the processes underlying the relationships regarding how air pollutants may enter the brain through BBB disintegration, nasal epithelium, and pulmonary vascular malfunction, causing negative neuroinflammatory and immunological responses [125]. Although most recent research on the effects of ambient pollution focused on respiratory disorders, these data point to a mechanism of how urban life can affect neural development that may affect intersecting neurotransmitters.

Natural environments improve human well-being since they serve as a source of rest and rejuvenation. Several studies demonstrate the positive effects of exposure to environmental scenery or their composite characteristics, such as plants and animals, including early childhood development, wellness, physical and mental health, emotional state, mortality, and resistance to disease [115,126]. Although studies are limited, a meta-analysis showed that outdoor physical activity boosts perceived energy and concentration while lowering unpleasant emotions such as anxiety, weariness, anger, and sadness [127]. Urban green spaces, and tree plantings, particularly those with significant biodiversity, benefit psychologically from exposure to nature.

It is noteworthy that poverty, a social disadvantage stratum, has definite associations with brain functions and mental stress of the pressures among people living in cities and the eagerness among minority people in rural areas to meet economic peers [128,129]. Numerous studies have demonstrated that poverty is linked to various environmental risk factors, including life stress, inadequate social support, and the absence of resources, including dietary needs or knowledge [130,131]. A recent study illustrated how mental health and brain markers of susceptibility are impacted by poverty. Early childhood poverty, measured at three months of age among children in an epidemiological group monitored from birth, revealed greater concentrations of conduct disorder symptoms during adolescence [132]. In neuroimaging studies, the orbitofrontal cortex, a crucial regulatory area involved in emotional transmission, revealed less volume in people exposed to early life poverty. In addition, the orbitofrontal cortex volume moderated the relationship between poverty and conduct disorder, pointing to a brain pathway including early trauma, impaired motivational and emotional control, and a higher risk of psychopathology [130,131].

### 2.3. Impact of Environment on Neurodegenerative Diseases

NDDs are related to aging, genetic, behavioral, environmental factors, and other neurological infections [133,134]. Environmental factors, such as heavy metals, pollution, poor diet, exposure to microorganisms, and lifestyle directly or indirectly impact brain health (Figure 1). Oluwole et al. reported that environmental work-related activities such as blacksmithing and drinking metal-contaminated well-water are possible sources of NDDs [9].

#### 2.3.1. In Utero Conditions

The human fetus is at risk of being exposed to chronic placental insufficiency (CPI), hypoxia, heavy metals, or hormonal disturbances during pregnancy. The growing fetus can be affected by the neurotoxins [40]. Exposure of the fetus in utero during pregnancy may likely substantially impact the future health of the born individual [135,136]. Studies have demonstrated that CPI or impairment in the umbilical cord may lead to umbilicus occlusion in fetuses, potentially causing synaptic dysfunction in neonates and neurodegeneration in their late ages [137,138]. During pregnancy and after childbirth, hormonal imbalances could harm the fetus’s mental development [139,140,141]. Hormonal levels in the fetus may get disrupted if the placental barrier between the mother and the fetus is compromised during pregnancy. Stress in pregnant mothers could also elevate glucocorticoid levels in the placenta and negatively influence the fetus by programming the hypothalamus-pituitary-adrenal (HPA) axis due to a change in the number and affinity of glucocorticoid receptors in the fetus [40,142,143]. There is limited human-based research on the effects of gestational or *in-utero* exposure on neurodegeneration to understand what pregnancy trimester could expose such disorder and the degree of impact on the fetus. Radioactive studies demonstrated that high levels of mercury and lead in umbilical cord blood due to prenatal exposures impair visual processing [144,145,146].

#### 2.3.2. Exposure to Metals and Pesticides

Toxic contaminants that abound in the environment are collectively known as heavy metals. These substances are widely diffused in the atmosphere, and their accumulation is elucidated to poison the ecosystem. Interestingly, the BBB’s primary function is to abort the crossing of heavy metals from the systemic circulation into the brain [147,148,149,150]. The formulation of metal-protein complexes renders metals unavailable to exert harmful effects and protect mature tissue against metal poisoning. Bibi et al. [151] and Jaishankar et al. [152] reported that the pathogenesis of NDDs is related to metal exposures such as lead, aluminum, zinc, iron, copper, and mercury from the soil, water, and other exposures. However, the available research findings are inconsistent in certain aspects, and more studies are required to link neuro metals and neurodegenerations. For example, some studies have linked high levels of aluminum in drinking water to an increased risk of AD, while others have shown no such link [153,154]. Furthermore, radioactive materials in contaminated areas revealed the movement of carbamate from the cornea to the retina of the eyes via aqueous humor, supporting the assumptions that radionuclides and pesticide exposure might be involved in the ocular route [155].

Environmental conditions contribute to cognitive impairment, motor abnormalities, disrupted mental states, neuroendocrine modifications, and sensory deficiencies [156,157], all possible symptoms of neurotoxicant exposure. Toxicants that interfere with energy generation have neuro-catastrophic effects on the CNS and PNS. The CNS is the primary target organ, with neurological effects right away seconds after injection of a toxicant into a body [156,158,159,160]. For instance, cyanide, a highly hazardous natural gas, is a famous example of a neurotoxicant, and interestingly, bacteria, fungi, algae, and some plants synthesize this chemical. Cyanide is a well-known cytochrome oxidase inhibitor (mitochondrial complex IV). The toxicant’s significant and neurological effects could block cellular respiration and limit neuronal ability sensitivity and immune response.

#### 2.3.3. Chemo Transmission in the Extracellular Environment

Discharging chemical signals through neurotransmission is responsible for communication between cells in the nervous system. Neurotransmitter release, reuptake, and metabolism are all tightly controlled processes. Neurotransmitter system disruptions can alter neuronal function and lead to neurodegeneration [161,162,163,164]. The neurotransmitter effect’s highly reactive and unpredictable features indicate their role in the etiology of NDDs, particularly DA-related diseases. Toxicants may also use transporters to access neurons where the native neurotransmitter has strong structural similarities; for example, 1-Methyl-4-phenylpyridinium ion entrance via the DA transporter [165]. Humans are most commonly exposed to other essential pollutants or toxins: pesticides and agrochemicals. Ball et al. [1] and Srivastav et al. [134] highlighted heavy metals ingestion, pesticides, and illicit drugs associated with NDDs.

Therefore, pesticide use and ingestion, among other toxic environmental pollutants, are a significant concern for human health. Insecticides, herbicides, and fungicides are among the most used agrochemicals. Organochlorines and carbamates are increasingly employed insecticides globally and may enter the human body via the respiratory tract, gastrointestinal tract, cutaneous contact, and water sources [166,167]. Although not a typical route of exposure, ocular exposure can also occur when pesticides accidentally splash into the eyes or when hands come into contact with the eye allowing pesticides to pass through the ocular tissue into the bloodstream [168]. Individuals working in agricultural areas, such as farmers and farm workers, are more likely to be exposed to pesticides. Notably, this group of individuals is exposed to these chemicals directly, while others are at risk of food contamination [169]. Acquired contaminants through ingestion and inhalation build up in the body, thus causing progressively destructive gene expression profiles of exposed soft tissues. Pesticides are among the leading pollutants that might modify the regulatory system, resulting in disease initiation and progression via epigenetic alterations [170]. Neuronal loss is the principal cause of cognition decline and motor dysfunction, and it has been linked to pesticide exposure [171], thus proving the implication of pesticide-related neurodegeneration. Similarly, laboratories and using chemicals and solvents could be a potential source of ingestion and deposit of neurotoxic substances in the body.

### 2.4. Impact of Climate on Neurodegenerative Diseases

Climate change results from gradual changes in weather compositions manifesting through rising temperatures, heat waves, floods, tornadoes, hurricanes, droughts, fires, loss of forests, glaciers, and the disappearance of rivers and desertification. These natural disasters can cause human pathophysiological diseases, including NDDs (Figure 1). Generally, climate change affects the entire biological system, and the impact is observed in all life cycles. Climate change also affects sensory stimuli and responsiveness and causes harm to the cognitive systems and the brain. Increasing high temperature due to global warming is one of the significant climatic conditions linked to NDDs. High temperatures affect cognitive ability, productivity, and working performance [172,173]. High environmental temperature conditions are substantial for dementia patients and could increase hospitalization rates, agitations, and mortality among PD patients [1,174]. High temperature and heat stress could lead to heat stress, oxidative stress, and excitotoxicity, especially among the elderly with thermo-compromised body regulations and neuroinflammatory events [175,176,177,178]. Habibi et al. reported that heat stress is known to have a degenerative effect on neurons [179]. Several studies on heat stress have been observed in neurons and their axons, glia, and cerebral vascular endothelium [180].

Neuropathological diseases have been linked to heat stress to cause heatstroke, dementia, stroke, migraine, and seizures [3,181]. A study on experimental animals on heat stress reported that heat stress compromised the BBB with mild traumatic brain injury to hyperthermia and long-term memory and learning deficits [182,183]. In developing countries, persistent heat stress caused by climatic changes might be crucial in increasing NDDs, and the severity could be higher [178,179]. Other diseases associated with climatic change include many re-emerging diseases and viruses such as Dengue, Zika, Chikungunya, West Nile, and Yellow Fever, which have direct neurological implications. For instance, Dengue infection is a tropical sickness with neurological complications in roughly 20% of infected patients, including encephalitis and encephalopathy [184,185]. Yellow fever can also cause catastrophic neurological problems, including encephalitis, acute neuroinflammation, and extensive neuronal damage [186]. West Nile is a neurotropic virus that can cause severe encephalitis in humans and horses. It is hazardous to babies, the elderly, and individuals with immune-compromised systems [187]. Zika infection during pregnancy leads to malformed brain development, microcephaly, cortical thinness, and blindness. This Zika virus can also cause meningoencephalitis in adults and develop into Guillain-Barre syndrome [188].

Human practices such as air pollution, traffic vehicle combustion, bush burning, and industrial pollutants contribute to invisible pollutants and neurotoxins in the atmosphere. Their deposit on food and water affects the nervous system [189]. In children, this may result in delays in cognitive development and lead to autism [190]. Particles from air pollution and occupational exposure can negatively affect the brain, potentially related to AD and PD [191,192]. Therefore, a healthier environment, increasing and maintaining trees in urban areas, green spaces, and green technology can improve the integrity of the environment and atmosphere and reduce the ingestion of toxic pollutants directly or indirectly. Hence, minimize neuronal hormone depletion and improve healthy well-being for all biogeographical cycles.

## 3. Managing Environmental and Climatic Influences in NDDs

### 3.1. Pharmacological Approaches

As controlling environmental and climatic factors that affect the neurotransmitters’ stability and promote the progression of NDDs is not always feasible, the involvement of pharmacological therapeutics in disease management is crucial. For innovative therapeutics, one of the successful strategies to alter the disease severity is the identification of therapeutic targets. Modulating the normal physiology and function of the CNS could be achieved by controlling the signal pathways of these neurotransmitters, hence, recognizing the mechanisms implemented in altered gene expression during CNS diseases. Understanding these mechanisms may improve the therapeutic efficacy or delay the disease onset.

Several studies have explored how environmental factors could employ their harmful effects on the CNS [40]. Lead is demonstrated to accumulate Aβ in brain tissue by reducing the efficacy of neprilysin (NEP) and insulin-degrading enzyme (IDE); both have a role in the degradation of Aβ [193,194]. The administration of NEP and IDE exogenously could be an excellent pharmacological approach to prevent lead-inducing toxicity. Oxidative stress is proposed as an additional environmental toxin that promotes NDDs, as evidenced by AD and PD-related reports [195,196]. The oxidative stress is induced because the heavy metals act as electron donors or acceptors, producing ROS [197]. Thus, using antioxidants in the therapeutic plan could facilitate disease management via the intoxication of heavy metals, thus, facilitating oxidative stress. It has been demonstrated that vitamin E and α-lipoic acid are essential in preventing copper-inducing neurotoxicity due to their antioxidant properties [198]. The natural bioactive compounds can also treat NDDs and avoid rapid disease progression owing to their antioxidant and polyphenolic activities. For instance, *Allium cepa* herbal extract from onion and Lutein extract was reported to diminish oxidative stress’s adverse effects [199,200]. Nutritional foods and the consumption of vegetables, medicinal herbs, herbal teas, spices, and other natural organic products should be included in daily diets as promising supplements to boost neuronal hormones. Such supplements contain polyphenols, a good source of antioxidants, enhancing cognitive function, defending the body, and improving mental well-being.

To reverse the effect elevated in NDDs, remedial steps can be performed to reduce the resulting cause. For example, the signaling pathways extracellular signal-regulated kinases (ERK_1/2_) and serine/threonine kinase are disrupted following metal exposure and drug abuse leading to impaired vesicular monoamine transporter (VMAT) proteins and DA active transporter (DAT) regulation [201]. Interestingly, similar effects on VMAT and DAT have been observed with striatum toxicity in manganese [202]. The administration of such substances with neuroprotective properties that can keep the signaling pathway and the expression level of VMAT and DAT proteins in the normal range is essential. It has been reported that ‘Zhen Wu Tang’ (ZWT), a traditional Chinese medicine, has been proven to ease neurodegeneration progression via maintaining the VMAT/DAT mRNA levels [203].

Moreover, Trolox has been demonstrated to reverse the harmful effect of manganese on the ERK _1/2_ pathway [201]. Hence, ZWT and Trolox can be pharmacological approaches to managing neuro-metal toxicity. Likewise, oxytocin can relieve the neurotoxicity induced by rotenone by inhibiting the expression of different caspases responsible for the cell apoptosis [204]. Similarly, in PD patients, targeting the elevated level of peroxisome proliferator-activated receptor coactivator-1α has a significant therapeutic impact on preventing dopaminergic neuron loss [205].

The NMDA receptor is an ion-channel receptor belonging to the glutamate receptors family that responds to the neurotransmitter glutamate and is found at most excitatory synapses. The induced toxicity following exposure to lead was reported to inhibit the expression of the NR2A subunit of the NMDA receptor required to maintain cognitive ability [206]. Taking taurine as a neuroprotective agent has been demonstrated to be effective in NMDA receptor malfunction and calcium influx-associated diseases by decreasing the calcium overload [207].

The body’s protein homeostasis is typically maintained by the degradation of accumulated, malfunctioned, and misfolded proteins via the Ubiquitin Proteosome Complex (UPC). The inhibition of UPC could lead to neurotoxicity owing to the deposition and aggregation of these proteins in the CNS [208]. Histone deacetylase (HDAC) plays a significant role in controlling histone modification in the cell, which is necessary to regulate the gene expression [209]. Using phenylbutyrate, trichostatin, and valproic acid as HDAC inhibitors have been reported to potentiate neuroprotective effects by controlling neurotrophic factors, such as reducing inflammation of brain-derived neurotrophic factor, neuronal death, and glial-derived neurotrophic factor [210,211]. Therefore, preventing the progression of NDDs caused by environmental hazards using pharmacological intervention can be one of the most successful approaches that help in neuroprotection.

The substantial climatic changes caused by increasing temperatures, heat waves, the disappearance of rivers, and desertification can promote the progression of NDDs, as described previously in the pathogenesis sections. Most of the currently used NDDs therapeutics were usually not affecting the disease progression but only the disease symptoms. For instance, levodopa and acetylcholine esterase inhibitors were demonstrated to relieve the symptoms in PD and AD, respectively [212]. One of the recent potential therapeutic approaches for NDDs is targeting misfolded proteins or preventing abnormal protein aggregation [213]. Moreover, the inhibition of different amyloidogenic proteins aggregation pathways using tetracycline nanoparticles, molecular chaperones, or antibodies is considered an effective therapeutic strategy to inhibit amyloid fibril formation, therefore delaying the NDDs related symptoms [214]. Hence, managing the symptoms related to the NDDs is the conventional therapeutic approach currently followed in the clinic, as controlling climatic factors is not always feasible.

### 3.2. Non-Pharmacological Approaches

Non-pharmacological interventions for NDDs could improve functionality or structural aspects of the neuroplasticity [215]. The main limitation of pharmacological components for NDDs is the difficulty associated with such elements crossing the BBB, which does not apply to most non-pharmacological approaches [216].

Various non-invasive techniques have been used as neuro-stimulators and modulators for the *CNS and PNS*, such as transcranial magnetic stimulation (TMS), transcranial direct current stimulation (tDCS), and transcranial alternating current stimulation (tACS). In the case of TMS, a magnetic field can be applied to the brain in different pulse forms, single, double, or multiple, which is approved by the Food and Drug Administration (FDA) to improve specific NDDs [217]. For instance, in the case of AD, a clinical trial involving 14 patients using high-frequency TMS showed improvement in the patient’s recall memory as assessed by cognitive tests [218]. On the other hand, using an electric technique such as tDCS or tACS on the scalp can reduce the effect of mental diseases [219]. The main difference between the two techniques relates to function: the tACS targets the brain *oscillations using a* sinusoidal current. At the same time, tDCS affects the cerebral cortex [220]. Other non-invasive techniques such as MRI-guided-focused ultrasound could help to open the BBB for mediating nanoparticles [221] or the *adeno*-associated viral vector [222] in rodent PD models.

Traditional Chinese therapies, such as acupuncture, herbs, and massage, have been investigated in pre-clinical and clinical trials as potential therapies for NDDs [223]. For example, Hwang et al. used combined acupuncture and Chunggan extract on a PD-induced mouse model, which improved motor function as assessed by different tests such as the rotarod test and the cylinder test [224]. In another study, the application of Tui Na massage on nerve-injured patients enhanced the recovery of damaged nerves and reduced the patients’ symptoms [225].

Stimulating neuroplasticity can be induced by phototherapy by exposing the patients to daylight or controlled artificial light and improving the patient’s rehabilitation. Despite the mechanism of this kind of therapy not being fully understood, it has been shown to delay NDDs progression and reduce nerve cell death [226]. Willis, Boda, and Freelance conducted a clinical phototherapy trial on 30 PD patients exposed to polychromatic light, red light, or discontinued polychromatic light for two weeks. Continuous exposure to polychromatic light was the only technique shown to be promising for the patients’ motor symptoms [227].

Evidence about the benefits of certain forms of physical exercise and nutrition has been proven effective in numerous neurological diseases [228]. Physical exercise (PE) could be beneficial concerning NDDs, such as PD [229,230] and AD [231,232], in terms of slowing the progression of the disease and improving life quality. Furthermore, PE can have a role to play in modulating epigenetic changes, for example, DNA methylation, histone modification, and non-coding RNA related to NDDs [233].

A balanced healthy diet is recommended for patients suffering from NDDs. For instance, a randomized clinical trial related to patients with PD who had been on a Mediterranean diet for ten weeks showed improvement in the patient’s memory and cognitive tests compared to a control group [234]. Vitamins or dietary supplements intake could also benefit NDD patients. For example, vitamin E was proposed as essential supplement protection of *dopaminergic* neurons in a PD animal model due to its antioxidant and anti-inflammatory activities [235].

## 4. Conclusion and Future Perspectives

This review highlighted the association between climate and environmental influences on NDDs. However, the link between early childhood lifestyle, emotional stability, amount of PE, environmental exposure, and the potential contributing factors, particularly neurotransmitters, to NDDs has not been adequately studied and, thus, needs further investigation. The role of environment and climate factors in neuronal hormone depletion and modulating effects on neurotransmitter systems are also limited. Taking cognizant of new instrumentations, review of previous methodological approaches and potential therapeutic targets on the gene, thermal stability, precise temperature inference, stress, high thermal variation due to global warming, and biochemical functions could offer a deepened knowledge of how much environment and climate are contributing to neuronal depletion and NDDs at large. The interconnecting variation of several neurohormones to age, gender, race, and stimulus-response of different ages, including in children and their neurochemical messengers, should increase from different perspectives.

In this work, climatic variations, such as temperature, heat stress, and heavy metal exposure and their interactions, are identified as key contributing factors that reduce the amount of ACh and DA in the brain, thus leading to NDDs. Some compounds such as mercury, copper, arsenic, manganese, lead, pesticides, biotoxins, persistent organic pollutants, and disruptive endocrine chemicals are possible agents of neuronal hormone depletion. Further studies are required to understand the impact of those compounds and how much they contribute to neuronal depletion. Oxidative stress and inflammation are related to aging and contribute to behavioral and metabolic fluctuations, mental stress from daily activities, and loss of several neurotransmitters leading to significant depression and anxiety disorders.

Environmental work-related activities such as crude extraction of mineral materials like gold and cobalt and drinking metal-contaminated water directly from wells are possible causes of NDDs. Other unsafe human practices, including cross-contamination of polluted agro-allied sites and radioactive materials in contaminated areas, could be other sources of NDDs prevalence.

Cell studies, animal studies, and enzymatic approaches, together with several aging neurological models’ combinations, would guide how neurotransmitters tracing at the genetic levels and how their contacts and the intracellular signaling for suppression are standard in a conducive environment. Elucidating these neurotransmitters and the number of neurotransmitters released during different conditions could reveal to what extent environment and climatic factors affect neurotransmitters and the possible influence on the production or depletion of neuronal hormones. Further investigation on the onset of pathologies of NDDs and the link with depression and other pathophysiological illnesses, seasonal trigger factors, inflammation, and anti-inflammatory factors that restrict cerebral immune responses could also be of great significance to neurodegenerative studies.

In the prospect of NDDs management, the implication of pharmacological and non-pharmacological approaches shall focus on the complete cure or long-term relief of the symptoms. One of the future strategies of NDDs management is inhibiting disease-associated protein deposition that promotes disease progression, such as using heat shock protein 104 (Hsp104) molecular chaperon. This approach showed neuroprotective effects by eliminating different amyloid conformations, reducing deposits of pre-amyloid oligomers, and precise Aβ depositions [236]. Several other therapeutic approaches are still under investigation. They could be implemented clinically soon to control the progression of NDDs, such as neuro-immunomodulator therapies, autophagy, targeting oxidoreductases, treatment with neurotrophic factors, and insulin therapy.

## Figures and Tables

**Figure 1 ijerph-19-12495-f001:**
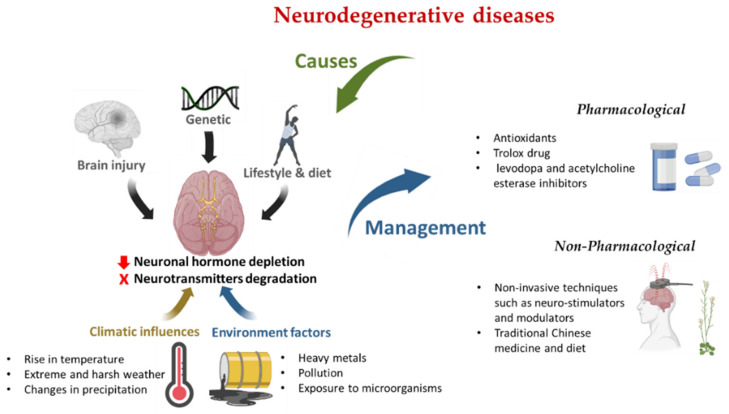
Summary of environment and climatic influences and approaches to manage NDDs Created by Biorender.com.

**Table 1 ijerph-19-12495-t001:** Summary of some neuronal hormones and neurotransmitters associated with key NDDs.

NDDs	Neurological Hormones/ Neurotransmitters	Agonist	Antagonist	Precursors	Metabolism	Biosynthesis	Receptors	Modulator	References
**Alzheimer** **Parkinson’s** **Dementia**	Acetylcholine	Nicotine Muscarine Cholinesterase inhibitors	Tubocurarine Atropine	Choline Acetyl-CoA	Choline acetyltransferase	Acetylcholinesterase/Role in neuronal activation of the adrenergic system	Nicotinic Muscarinic	Galantamine, Rivastigmine, Donepezil	[74,75,76]
	Dopamine	Apomorphine Amphetamine	Neuroleptics Domperidone	Phenylalanine Tyrosine L-DOPA	Monoamine oxidases (MAO) Catechol-O-methyl transferase (COMT)	DOPA decarboxylase/ DA converges to modulate nuclear factor erythroid 2-like2 (Nrf2)	G protein-coupled receptors/ D1-D5 TAAR	Bromocriptine, Pergolide, lisuride, Cabergoline, Ropinirole, Pramipexole Safinamide	[77,78,79]
	Serotonin	Azapirones Triptans	Trazodone Nefazodone	5-HTP Cation channel	MAO	Aromatic L-amino acid decarboxylase Anxiolytics: buspirone MAOI: Furazolidone, Isocarboxazid, linezolid, phenelzine, selegiline, and tranylcypromine	G protein-coupled receptors 5-HT1, 5-HT2, 5-HT3, 5-HT4, 5-HT5, 5-HT6, 5-HT7	Citalopram, Escitalopram, Fluoxetine, Paroxetine, Sertraline (Zoloft), Clemizole, Locaserin, Trazodone and Fenfluramine	[80,81,82]
	Melatonin	Ramelteon	Luzindole	Serotonin (5-hydroxytryptamine);N-acetylserotonizn		6-hydroxymelatonin (6-HMEL	MT1 MT2 receptors	Meladox Melatin Melatonin	[83,84]
	Histamine	2-Pyridylethylamine Betahistine Histamine, HTMT, L-Histidine, UR-AK49	4-Methyldiphenhydramine, Alimemazine Antazoline Azatadine	Histidine	Diamine oxidase (DAO) Histamine-N-methyltransferase (HNMT)	Decarboxylation of L-histidine by histidine decarboxylase (HDC)	H1–H4 receptors	Cetirizine, Fexofenadine, Cyproheptadine, fish, cheese, fermented meat, beverages, and vegetables	[85]
	Epinephrine	Amidephrine Anisodamine Anisodine Cirazoline	Propranolol Phentolamine	Dopamine	MAO COMT	Catecholamine pathway Phenylethanolamine N-methyltransferase	Adrenergic (α and β) receptors	Micronefrin, Nephron, VapoNefrin	[85]
	Norepinephrine	Sympathomimetic drugs Clonidine Isoprenaline Antagonists	Tricyclic antidepressants Beta-blockers Antipsychotics	Dopamine	MAO-A COMT	Dopamine β-monooxygenase	Adrenergic (α and β) receptors	Atomoxetine, Reboxetine, viloxazine	[86]
	Thyroid hormones	Sobetirome (GC-1)	7 transmembrane-spanning receptor (7TMR)	T0–T4	Glucuronidation, sulfation and deiodination (type 1 (D1), type 2 (D2) and type 3 (D3) iodothyronine deiodinases	The interface of the apical thyroid cell plasma membrane and the colloid, and TG molecules containing T4 and T3 are stored in the follicle lumen	THRA1, THRA2, ERBA1	L-thyroxin (T4); 3, 3′,5-triiodo-L-thyronine (T3)	[87,88]
	Tyrosine	Amidephrine Anisodine Buspirone Cirazoline	Axitinib Dasatinib Erlotinib Imatinib Nilotinib	Alkaloid (Morphine; p-coumaric acid) Pigments (melanin)	L-DOPA via tyrosine hydroxylase (TH)	Phenylalanine by phenylalanine hydroxylase	Insulin receptor, Vascular endothelial growth factor Fibroblast growth factor Platelet-derived growth factor receptor	L-tyrosine, Levothroid, Levoxyl, Synthroid	[89,90]
**Encephalopathy** **Parkinson’s Huntington’s**	Glutamate	Kainic acid	N-methyl-d-aspartate (NMDA)	N-acetyl-L-glutamate δ-1-Pyrroline-5-carboxylate β-citryl glutamate L-γ-glutamyl-L-cysteine	Glutamate to N-acetyl-L-glutamate by N-acetyl glutamate synthase (NAGS) and triggered by L-arginine by succinate, CoA, N-acetyl-L-aspartate N-acetyl-L-Glutamate γ –aminobutyrate Glutathiones	L-proline L-arginine	α-amino-3-hydroxy-5-methyl-4-isoxazole propionic acid receptor (AMPA) Kainic acid receptors; N-methyl-D-aspartate receptor (NMDA)	GLYX-13, NRX-1074, CERC-301 (MK-0657), Esketamine (Ketanest S) Lanicemine.	[91,92,93]
	Gamma-aminobutyric acid	Muscimol Baclofen	Bicuculline Gabazine Flumazen	Glucose to Glutamate via TCA enzyme Pyruvate Glutamine	GABA transaminase converted to 4-amino butanoic acid (GABA) and α-ketoglutarate to succinic semialdehyde and Glutamate	Glutamate by glutamate decarboxylase (GAD) via pyridoxal phosphate	GABA (A-C)	Benzidiazepines e.g diazepam and alprazolam	[94,95]
	Glycine	β-alanine L- Serine D- Serine L-Proline Taurine	Brucine Caffeine Picrotoxin Strychnine	Glutathione Porphyrins Purines Haem Creatine	L-serine via glycine hydroxyl-methyltransferase	Serine Threonine Choline Hydroxyproline	Glycine receptors (GLRA1, GLRA2, GLRA3, GLRA4) and a single β-subunit (GLRB)	D-cycloserine, (R,S)-ketamine	[96,97]
**Alzheimer** **Parkinson** **Ischemia-reperfusion** **Anxiety**	Nitric oxide	S-Nitroso-N-acetyl-DL-penicillamine	3-Bromo-7-nitroimidazole N-N-dimethylarginine	L-arginine L-citrulline	Cysteine S-nitrosylation Thiol amino acids	NOS neuronal (nNOS) NOS inducible (iNOS) Endothelial NOS (eNOS)	Guanylyl cyclase	INomax	[98]
	Hydrogen sulfide	Trimebutine	Glibenclamide	Methanethiol Ethanethiol Thioglycolic acid.	Cystathionine γ-lyase (CSE) Cystathionine β-synthetase (CBS) 3-mercaptopyruvate sulfurtransferase (3-MST)	L-cysteine cystathionine γ-lyase Cystathionine β-synthase (CBS) 3-mercaptopyruvate-sulfurtransferase Aspartate Aminotransferase D-cysteine D-amino acid oxidase (DAO)	NMDA receptors		[99]

## Data Availability

The authors confirm that the data supporting the findings of this study are available within the article.

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
