# Peer review of "Neurodegenerative Diseases: Implications of Environmental and Climatic Influences on Neurotransmitters and Neuronal Hormones Activities"

_ijerph, 2022, doi:10.3390/ijerph191912495_

Round 1

Reviewer 1 Report (Previous Reviewer 2)

The topic is interesting, however, there are some critical major points to address:

1. in the introduction section, the authors should explain much better the aim of their work, mainly it is important to improve the rationale underlining the link with pharmacological and non-pharmacological approaches because it is not clear the connection between environment and climate with “disease management strategy”.

2. The authors should structure better the whole text to be clearer to readers:

2a. As the paragraphs 2.2, 2.3, 2.4 represent the core of the review, it should be better to change in such a way to create new paragraphs. For instance,

3. Impact of Environmental on Neurodegenerative diseases

4. Impact of climate on Neurodegenerative diseases

5. Implications of Environmental and climate influences on neurotransmitters and neuronal hormones activities

2b. Concerning the treatment approaches, the authors should be clearer on how presenting this paragraph. For instance, they could change the title in such a way to highlight how the different treatments can influence the environmental and climate factors: the authors reported literature on lead or manganese but what about the role of climate factors?

In the sentences from 422 to 430, the authors reported natural compounds but this must be moved in the next paragraph concerning the non-pharmacological approaches.

3. In the conclusion section, the authors could refer also to future developmental concerning the pharmacological and non-pharmacological approaches.  

4. I think that the Figure 1 is difficult to follow. The authors could modify it, and also explain the picture in the legend. For instance, below “environmental and climate influences”, there are only climate factors, whereas below “indirect impact” and “causes” there are environmental factors …  

Author Response

We thank the referees for their thoughtful and helpful comments. We respond to each point below; their suggestions have undoubtedly improved the paper, for which we are very grateful. 

Reviewer 1:

The topic is interesting, however, there are some critical major points to address:

  1. in the introduction section, the authors should explain much better the aim of their work, mainly it is important to improve the rationale underlining the link with pharmacological and non-pharmacological approaches because it is not clear the connection between environment and climate with “disease management strategy”.

Thank you for your comment. The below part has been added in the introduction section of the revised manuscript, and blue is highlighted (lines 114-121)

  1. The authors should structure better the whole text to be clearer to readers:

2a. As the paragraphs 2.2, 2.3, 2.4 represent the core of the review, it should be better to change in such a way to create new paragraphs. For instance,

  1. Impact of Environmental on Neurodegenerative diseases
  2. Impact of climate on Neurodegenerative diseases
  3. Implications of Environmental and climate influences on neurotransmitters and neuronal hormones activities

The structure of the manuscript has been amended according to the suggested structure. Thank you!

2b. Concerning the treatment approaches, the authors should be clearer on how presenting this paragraph. For instance, they could change the title in such a way to highlight how the different treatments can influence the environmental and climate factors: the authors reported literature on lead or manganese but what about the role of climate factors?

Thank you for your valuable suggestion. The title of the treatment section has been changed from (Pharmacological and Non-Pharmacological Approaches for Neurodegenerative Diseases Management) to (Managing Environmental and Climatic Influences in NDDs) in order to highlight the influences of different treatment approaches, and the new title has been blue-highlighted in the revised version of the manuscript. Regarding the role of these treatment approaches on climatic factors, the below paragraph has been added to the revised version of the manuscript and blue highlighted (line 499-512)

In the sentences from 422 to 430, the authors reported natural compounds but this must be moved in the next paragraph concerning the non-pharmacological approaches.

This has been amended accordingly.

  1. In the conclusion section, the authors could refer also to future developmental concerning the pharmacological and non-pharmacological approaches.

Thank you for your comment. This part has been added to the conclusion section in the revised manuscript, and blue highlighted (lines 611-621)

  1. I think that the Figure 1 is difficult to follow. The authors could modify it, and also explain the picture in the legend. For instance, below “environmental and climate influences”, there are only climate factors, whereas below “indirect impact” and “causes” there are environmental factors …

Thank you for your suggestion. We have improved Figure 1 accordingly. The Figure is just a line summary of the review, and we have updated it according to the above suggestion.

We also confirm that the manuscript has been proofread.

Reviewer 2 Report (New Reviewer)

1. The main criticism of this study is the lack of suitable conclusions and remarks that could account for future scientific application. Intense research is being conducted in many areas related to the etiology of neurodegenerative diseases, from basic science seeking the roots of the disease to therapy development to find effective treatments. However, translating such basic research data into the clinic has not been straightforward and has presented many scientific and regulatory challenges for scientists. In this regard, authors should interpret and discuss the significance of their findings in the specific section (conclusions and remarks).

2. Importantly, in the Figure1 authors summarize the most common neuronal hormones and neurotransmitters associated with NDDs distinguishing their effects in the pathogenesis of the specific disease is not possible. In the same way, the authors used general terms (NDDs) for describing Table 2. Also, in the body of the manuscript authors did not explain the issue and finding by separating the diseases. This shortage makes it very difficult to understand the content and follow the subjects.

3. What do the authors mean by environmental and climatic influences? How they separated these two categories from each other? In the introduction part, they talk more about NDDs rather than these kinds of effects that they presumed. They just mentioned a short general description regarding problems in the regulation of body temperature without any molecular evidence. They should explain the importance of their topic with stronger evidence.

4. They should add a table that shows the full details with the precise mechanism of action (if not possible with the proposed mechanism of action) of the environmental as well as climatic influences in each neurodegenerative disease.

Author Response

Response to referees

We thank the referees for their thoughtful and helpful comments. We respond to each point below; their suggestions have undoubtedly improved the paper, for which we are very grateful.  

Reviewer 2:

  1. The main criticism of this study is the lack of suitable conclusions and remarks that could account for future scientific application. Intense research is being conducted in many areas related to the etiology of neurodegenerative diseases, from basic science seeking the roots of the disease to therapy development to find effective treatments. However, translating such basic research data into the clinic has not been straightforward and has presented many scientific and regulatory challenges for scientists. In this regard, authors should interpret and discuss the significance of their findings in the specific section (conclusions and remarks).

Thank you, we have identified some of the significance of our findings in this manuscript's conclusion and future perspectives section.

  1. Importantly, in the Figure1 authors summarize the most common neuronal hormones and neurotransmitters associated with NDDs distinguishing their effects in the pathogenesis of the specific disease is not possible. In the same way, the authors used general terms (NDDs) for describing Table 2. Also, in the body of the manuscript authors did not explain the issue and finding by separating the diseases. This shortage makes it very difficult to understand the content and follow the subjects.

Thank you for your suggestion. We did not distinguish them from different diseases; what we did was list some NDDs, respectively. Meanwhile, we have corrected the text to capture the above concern. We have discussed several NDDs in lines 47-74.

  1. What do the authors mean by environmental and climatic influences? How they separated these two categories from each other? In the introduction part, they talk more about NDDs rather than these kinds of effects that they presumed. They just mentioned a short general description regarding problems in the regulation of body temperature without any molecular evidence. They should explain the importance of their topic with stronger evidence.

Thank you for your comments. Climatic influences can result in a rise in average and extreme temperature; extreme and harsh weather conditions, and changes in precipitations, while environmental factors can rise due to metal contaminations; pollution (air, soil and water); neuronal hormones depletion; neurotransmitters fluctuations and heat strokes which also summarized in Figure 1. We did not focus on the molecular evidence related to this paper. Meanwhile, we have carefully restructured the introduction section accordingly.  

  1. They should add a table that shows the full details with the precise mechanism of action (if not possible with the proposed mechanism of action) of the environmental as well as climatic influences in each neurodegenerative disease.

Thank you for the critical note. The aim of this work is to discuss the roles of different neuronal hormones and neurotransmitters in the pathogenesis of NDDs and highlights the influence of environmental and climatic factors on their activities. Our subsequent work will focus on some proposed mechanisms between environmental factors and climatic neurodegenerative diseases. We have identified this as future work in the conclusion and recommendation section.

We also confirm that the manuscript has been proofread.

Round 2

Reviewer 1 Report (Previous Reviewer 2)

The authors have adequately answered all my questions. The work is eligible for publication.

Reviewer 2 Report (New Reviewer)

The paper now would be more valuable for the target scientists.

This manuscript is a resubmission of an earlier submission. The following is a list of the peer review reports and author responses from that submission.

Round 1

Reviewer 1 Report

In the manuscript authors aimed to highlighted implications of environmental and climatic influences on neurotransmitters and neuronal hormones activities in development/risk/progression of NDD. Manuscript is very interesting but it needs some rearrangements to fulfil the main goal of this work.

I encourage authors to change the subtitles to : 1. Introduction, 2. Pathogenesis of NDD. 2.1. Role of neurotransmitters and neuronal hormones in pathogenesis of NDD, 2.2. Impact of Environment on Neurodegenerative Diseases, 2.3.. Impact of Climate on Neurodegenerative Diseases, 3. . Pharmacological and Non-Pharmacological Approaches for Neurodegenerative Diseases Management, 4. Conclusions and Future Perspectives.

Author Response

We thank the referee for their thoughtful and helpful comments. We respond to each point below; their suggestions have undoubtedly improved the paper, for which we are very grateful. Please note that all changes are in Red Font.

Thank you for your comments. We have re-arranged as suggested.

Reviewer 2 Report

In this review, the authors sought to discuss the role of major neuronal hormones and neurotransmitters in the pathogenesis of NDDs and the implications of environmental and climate influences on neurotransmitters and neuronal hormones activities, besides of pharmacological and nonpharmacological approaches.

The topic is interesting, however, there are some critical major points to address:

 - Figure 1 does not reflect what the authors describe in the Introduction section on page 1 (lines 39-43). Schizophrenia and depression are pathologies not linked to NDDs, and if yes the authors should specify. Moreover, in table 1 on page 9 (line 359) the authors should explain why to insert schizophrenia in the NDDs column .

 - The authors wrote at page 3 (lines 85-88) that : "this review discusses the roles of major neuronal hormones and neurotransmitters in the pathogenesis of NDDs and the implications of environmental and climate influences on neurotransmitters and neuronal hormones activities ....". However, this topic was not addressed by the authors. There is no evidence concerning the impact of environmental factors on neurotransmitters / hormones. If yes, the authors should explain better.

 - The authors wrote at page 3 (lines 88-89) that: "the pharmacological and non-pharmacological approaches that are common to reduce the susceptibility to NDDs are also discussed." In my opinion, this part of the work has nothing to do with the one described above. Are there any links between pharmacological and non-pharmacological approaches and environmental and climatic effects and neurotransmitters and hormones? If so, the authors should explain better. .

 - Figures 1 and 2 gave no further information and therefore do not seem necessary. However, if pertinent, the authors could insert a single figure that summarizes the whole review.

 - In tables 1, 2, 3, the authors could specify the modulation of each neurotransmitters/hormones in specific NDDs as reported in the text, adding a new column

 - In tables 2, 3 the authors should follow the same order of neurotransmitters/hormones as reported in the txt (For example: Tyrosine, Glutamate and GABA).

Author Response

We thank the referee for their thoughtful and helpful comments. We respond to each point below; their suggestions have undoubtedly improved the paper, for which we are very grateful. Please note that all changes are in Red Font.

 - Figure 1 does not reflect what the authors describe in the Introduction section on page 1 (lines 39-43). 

Response: Figure 1 was deleted and we have replaced it to show an overview of the points addressed in this manuscript.  Kindly see line 560.

- Schizophrenia and depression are pathologies not linked to NDDs, and if yes the authors should specify.

Thank you for your comment. Schizophrenia and depression are pathologies related to neurodegenerative diseases. Several studies have reported them to be linked to NDDs and as an early warning sign of psychological disorders and could degenerate brain damage.

- Moreover, in table 1 on page 9 (line 359) the authors should explain why to insert schizophrenia in the NDDs column.

We have changed schizophrenia to “psychological disorder” as they form behavioral disorder (mental retardation). Thank you.

- The authors wrote at page 3 (lines 85-88) that : "this review discusses the roles of major neuronal hormones and neurotransmitters in the pathogenesis of NDDs and the implications of environmental and climate influences on neurotransmitters and neuronal hormones activities ....". However, this topic was not addressed by the authors. There is no evidence concerning the impact of environmental factors on neurotransmitters / hormones. If yes, the authors should explain better.

We thank the reviewer for his valuable comment. This review is focusing on the effect of environmental and climatic influences on neurotransmitters and neuronal hormones activities on NDDs. The last part of the review has discussed the pharmacological and non-pharmacological approaches that could improve the disease management and control NDDs progression. Apparently, there is an important link between the pharmacological and non-pharmacological approaches, the environmental and climatic effects, and neurotransmitters and hormones, as the control of environmental and climatic factors that affect the neurotransmitters' stability or hormones' activity and promote the progression of NDDs is not always feasible. Therefore, the involvement of pharmacological therapeutic and non-pharmacological approaches in disease management is crucial. This has been explained in detail in the last part of this review 3) Pharmacological and Non-Pharmacological Approaches for Neurodegenerative Diseases Management. We further developed a new section to provide insights on the implications of environmental and climatic effects on neurotransmitters/hormones and summarised Table 4 (line 637-639). To the best of our knowledge, there is limited work in understanding how much environment and climate contribute to neurotransmitter depletion which we have suggested as a future perspective in this manuscript.

- Figures 1 and 2 gave no further information and therefore do not seem necessary. However, if pertinent, the authors could insert a single figure that summarizes the whole review.

Figures 1 and 2 have been deleted. We have developed a new Figure 1 that summarised the whole review. Kindly see line 560.

- In tables 1, 2, 3, the authors could specify the modulation of each neurotransmitters/hormones in specific NDDs as reported in the text, adding a new column.

This has been added and listed some modulators are being used in Tables 1-3. Thank you!

- In tables 2, 3 the authors should follow the same order of neurotransmitters/hormones as reported in the txt (For example: Tyrosine, Glutamate and GABA).

We have followed the order in the text for Tables 2 and 3 accordingly. Thank you so much.

Reviewer 3 Report

The manuscript entitled  “Neurodegenerative Diseases: Implications of Environmental  and Climatic Influences on Neurotransmitters and Neuronal Hormones Activities” by Ayeni et al., as the title suggests, should be dedicated to implications of neurotransmitters and neuronal hormones into neurodegenerative diseases development with the accent on the role of climate and environmental conditions on these influence.

The stated topic of the review is extremely relevant, this title implies a wide coverage of experimental data, and such an attractive title arouses genuine interest. 

Unfortunately current narrative is a list of brief characteristics of a number of most well-known neurotransmitters and neuronal Hormones and set of facts from experimental works ( table 1) presented sequentially, without any generalization or analysis.

Figure 2 represents the structure of some neuronal hormones and transmitters and (such details is not entirely justified within the framework of the stated topic) , fig. 1 and 3 are obvious and don't make much sense.

Even in the “Conclusions and Future Perspectives”, the authors did not bother to give at least a minimal generalization of the presented data, obtained by other authors, which is really interesting in the light of the relevance of the review.

The recommendations of the authors at the end of the review (Nutritional foods and consumption of vegetables, medicinal herbs, herbal teas, spices and other natural organic products should be included in the daily diet) look obvious even without reading this review.

Summing up, in the announced review the reader expects to find perspective analysis deepens our understanding in the area of neurodegenerative diseases from the point of implications of environmental and climatic influences on as well as professional discussion concerning the creation of new instruments or methodical approaches for potential therapies.  But these expectations are in vain. Unfortunately, it should be noted that this manuscript is a simple list of well-known facts, it does not contain any new significant ideas for future development of investigations or notional generalizations, and it is poorly structured. The review is weak and unimpressive; it is superficial in terms of analysis.

It should be concluded that current manuscript falls well below the required threshold for publication. 

Author Response

We thank the referee for their thoughtful and helpful comments. We respond to each point below; their suggestions have undoubtedly improved the paper, for which we are very grateful. Please note that all changes are in Red Font.

- The stated topic of the review is extremely relevant, this title implies a wide coverage of experimental data, and such an attractive title arouses genuine interest. Unfortunately, current narrative is a list of brief characteristics of a number of most well-known neurotransmitters and neuronal Hormones and set of facts from experimental works (table 1) presented sequentially, without any generalization or analysis.

Thank you for your comments. We have added a new section and presented some analysis of some of the work done in this regard (see Table 4). We also discussed some perspectives and sociological analyses of how environment and climate affect brain health including in children (lines 421-559). Unfortunately, there is limited information on how these contributing factors (environment and climate impact neurotransmitters/ neuronal hormones depleted). We have suggested this as future perspectives and as recommendations for future work if there are any.

- Figure 2 represents the structure of some neuronal hormones and transmitters and (such details is not entirely justified within the framework of the stated topic) , fig. 1 and 3 are obvious and don't make much sense.

Thank you for your comment, we have replaced these figures and re-summarized the figures to capture the idea of the whole manuscript. Kindly see the new Figure 1 (line 560).

- Even in the “Conclusions and Future Perspectives”, the authors did not bother to give at least a minimal generalization of the presented data, obtained by other authors, which is really interesting in the light of the relevance of the review.The recommendations of the authors at the end of the review (Nutritional foods and consumption of vegetables, medicinal herbs, herbal teas, spices and other natural organic products should be included in the daily diet) look obvious even without reading this review.

Thank you for your comments, we have improved on the conclusion and future perspectives as their limited studies on this area of interest. So, we have identified some gaps to be filled on how new models could be useful to understand the exact amount of neuronal depletion and neurotransmitters degradation in relation to environment and climate and how other factors affect their performance.

- Summing up, in the announced review the reader expects to find perspective analysis deepens our understanding in the area of neurodegenerative diseases from the point of implications of environmental and climatic influences on as well as professional discussion concerning the creation of new instruments or methodical approaches for potential therapies. But these expectations are in vain. Unfortunately, it should be noted that this manuscript is a simple list of well-known facts, it does not contain any new significant ideas for future development of investigations or notional generalizations, and it is poorly structured. The review is weak and unimpressive; it is superficial in terms of analysis. It should be concluded that current manuscript falls well below the required threshold for publication. 

Thank you for your comments. We have further overhauled the manuscript based on your own suggestions and other reviewers’ recommendations, kindly see line (841-865). We believed that this work will serve as a basis and open up further research to understand if there are contributing factors such as climate and environment on different neurotransmitters and how much neurohormonal depletion is affected by these factors. Overall, we have further reviewed the flow, and logic and also made some significant contributions to the synthesis of some articles related to neuronal hormones towards managing neurodegenerative diseases.

Round 2

Reviewer 2 Report

The authors replied to my comments.

However, I still have doubts about the homogeneity and connection between the various paragraphs.

Furthermore:

1. the authors should again explain better the aim of their work, starting from what's lacking in the literature and mainly how they would like to structure their review. In the lines 90-93 page 2, the authors did not report the pharmacological and non phramacological approaches as further aim of the work. When the authors will introduce the structure of the review, they should explain better the meaning of the different tables.

2. to be clearer, the authors should add a title for each paragraph related to the tables 1, 2, 3, 4. In the paragraphs related to 2.1.3. Serotonin and  2.1.4 Melatonin, the authors introduced queekly the Table 4 .......What's the meaning of the Table 4?

3. be careful to: 1. acronymous (for instance lines 149, 423, 429), Enghish errors (for instance line 461 Environment have a positive or negative...)